# Facing a blind alley - Experiences of stress-related exhaustion: a qualitative study

Sara Alsén ,[1,2] Lilas Ali ,[1,2,3] Inger Ekman,[1,2] Andreas Fors [1,2,4]

¹Institute of Health and Care Sciences, Sahlgrenska Academy, University of Gothenburg, Gothenburg, Sweden
²Centre for Person-Centred Care (GPCC), University of Gothenburg, Gothenburg, Sweden
³Psychiatric department, Sahlgrenska University Hospital, Gothenburg, Sweden
⁴Research and Development Primary Health Care, Region Västra Götaland, Gothenburg, Sweden

**Correspondence to**
Sara Alsén; sara.alsen@gu.se

## ABSTRACT

**Introduction** Mental illness is a major concern in many countries. In Sweden, stress-related mental illness is currently the most frequent reason for sick leave.

**Objective** This study aimed to explore patients' experiences of stress-related exhaustion.

**Design** A qualitative study with interview data analysed using a phenomenological hermeneutic method.

**Setting** Participants were selected from public primary healthcare centres in a larger city in western Sweden.

**Participants** Seven women and five men on sick leave from work due to stress-related exhaustion were included in the study.

**Findings** Stress-related exhaustion was experienced as a loss of access to oneself and one's context and feelings of being trapped and lost in life. The condition had a significant impact on personal identity, raised existential issues and was interpreted as facing a blind alley. Participants described a mistaken direction in life, being forced to stop, change direction and act differently.

**Conclusion** Stress-related exhaustion is a challenging, life-changing existential experience that involves a crisis with an opportunity for new insights. Careful consideration of patients' narratives together with the expertise of healthcare professionals can be combined to improve health and optimise recovery based on individual's situation.

## Strengths and limitations of this study

► This study explored a less studied area, patient's experiences of stress-related exhaustion in the early stages of sick leave from work.
► A sample of women and men of various ages was recruited to ensure representation.
► The finding gave insight into the meaning of living with stress-related exhaustion.
► The study included 12 participants' narratives, which could be argued do not apply to other people's lives in different contexts.

## INTRODUCTION

Mental illness is a major concern in many countries,[1–4] and the days of work-related sick leave related to mental illness cause increased psychosocial stress.[5 6] In Sweden, the number of sick-leave days related to mental illness has more than doubled over the last decades. Stress-related mental illness is the most frequent reason for sick leave,[7 8] particularly in the human service sector.[9] Besides the negative impact on the patients' well-being and risk of stigmatisation,[10] its costs to society are high.[8] Research has found that exposure to stress increases the risk of both physical[11 12] and mental[13 14] symptoms, and cognitive impairments are frequently reported.[15] Distress has been shown to increase with unsupportive encounters.[16 17] Internationally, the term 'burnout' is used

to describe the consequences of severe and long-term stress that is related to psychosocial factors at work.[11] The three key dimensions of burnout are overwhelming exhaustion, feeling detached from work ('cynicism') and diminishing efficiency at work. The definition of burnout has broadened beyond the core dimensions, and exhaustion and disengagement are also included along with non-work-related factors such as caregiver burden, work–family conflict and unemployment.[18 19] Burnout, however, is not considered applicable in clinical practice.[20 21]

In 2003, the clinical diagnosis 'exhaustion disorder' was introduced in the Swedish version of the International Classification of Diseases Version 10 (ICD-10) with the diagnostic code F43.8A to describe patients with stress-related exhaustion. The diagnostic criteria of at least one identified stressor, work-related or non-work-related, should have been present for at least 6 months and that the clinical picture is dominated by a lack of psychological energy. Four of the following symptoms should be present almost everyday for at least 2 weeks: concentration or memory impairment, emotional instability, reduced ability to cope with demands and/or time pressure, disturbed sleep, apparent physical weakness and physical symptoms such as muscular pain.[22] Previous studies in patients with burnout have found that the

condition may result from moral dilemmas and power-lessness in relation to work particularly in the context of healthcare.[23 24]The lead-in to burnout is characterised by symptoms such as headache, muscular pain and fatigue and difficulties in living up to the emotional demands of family and friends. Self-image is threatened, and a need to refrain from social activities causes feelings of guilt.[25] Patients describe a situation in which their ideal self is not matched by reality.[26 27]Underlying factors of stress-related exhaustion are mainly associated not only with work demand but also with relational factors in private life or a lack of work/life balance.[28–30]

Previous research has mainly focused on the triggers for burnout or addressed specific occupational groups such as healthcare professionals. However, stress-related exhaustion is increasing in other demographic groups, and an improved understanding of how the condition affects the everyday life of patients on sick leave is important to reduce suffering and enhance recovery. This study aimed to explore patients' experiences of stress-related exhaustion.

## METHOD
### Design
An explorative qualitative interview study, the data were analysed using phenomenological hermeneutics.[31] This approach is designed to develop deeper understanding of individuals' perspectives by interpreting their narratives as texts. The present study is based on the Standards for Reporting Qualitative Research guidelines.[32]

### Participants and setting
Participants were selected from public primary healthcare centres in a large city in western Sweden. Designated healthcare professionals screened patients on sick leave due to stress-related exhaustion from medical records. After patients consent, the first author contacted the patients by telephone to schedule a time for an interview. A representative sample was recruited to ensure a variety of men and women of various ages. Participants were invited to participate based on the following inclusion criteria: (a) diagnosed with exhaustion disorder by a physician, (b) ongoing sick leave no longer than 6 months due to exhaustion disorder and (c) having a physical and mental capacity to participate in an interview. The interviews were conducted until the authors considered the research question as fully answered. Nineteen people were contacted and informed about the study; 3 declined and 4 withdrew before the interview due to lack of sufficient energy to participate. In total, 12 people were recruited, 7 women and 5 men, aged 25–46 years (table 1). Six of the participants were married and six were single, and their current percentage sick leave hours ranged between 50% and 100%, except for one who resumed full-time work 2 days before the interview.

### Data collection
The interviews were conducted between June 2018 and February 2019 by the first author. Each participant

| Table 1 | Characteristics of the participants |
|---|---|
| **Gender** | **Age** |
| 1: Male | 38 |
| 2: Female | 30 |
| 3: Female | 25 |
| 4: Female | 29 |
| 5: Female | 37 |
| 6: Female | 43 |
| 7: Female | 43 |
| 8: Male | 46 |
| 9: Female | 36 |
| 10: Male | 28 |
| 11: Male | 41 |
| 12: Male | 25 |

decided whether their interview took place at the first author's office, at a café or in the participant's home. The interviews were performed 2–4 months after sick leave had commenced and lasted between 28 min and 79 min (median 45 min) and were recorded and transcribed verbatim by the first author. The aim of the study was explained to the participants and they were asked to narrate their experiences. The opening question was 'How do you experience being affected by stress-related exhaustion?'. To obtain more in-depth narratives of the experiences, the interviewer asked probing questions such as 'What does that mean to you? How do you feel about that?'.

### Patient and public involvement
The study was conducted within the University of Gothenburg Centre for Person-Centred Care (GPCC), where patient and public involvement is essential. A patient representative was involved in the present project, which conducted research in the area of stress-related mental illness.

### Ethical considerations
The participants received verbal and written information about the study and assurance that they could withdraw at any time. All interview data were treated confidentially. Sharing lived experiences can be difficult and may raise sensitive issues. If required, patients were offered consultation with a counsellor. Written consent was obtained from each participant.

### Data analysis
The interviews were analysed using phenomenological hermeneutical method. The intention with this method is to generate an understanding of the meaning of a particular phenomenon, in this case, living with stress-related exhaustion. The method is inspired by Ricoeur's theory of interpretation,[31 33] which has been used in numerous studies during the last 20 years.[34 35]The analysis consists of three intertwined steps: naïve reading, structural analysis

and interpretation of the whole.[31] In the naïve reading, the text is read repeatedly to grasp its meaning as a whole. In the structural analysis, the text is divided into meaning units, which are abstracted and formed into subthemes. The structural analysis entails a dialectical movement shifting focus between the meaning units and the overall impression of the text. Several structural analyses and interpretations of the text were performed in this study. In the final part of the analysis; interpretation of the whole, the entire text was reread and interpreted in relation to the authors' preunderstandings, naïve reading, structural analyses, research related to the area and Jaspers philosophy[36] on limit situation to formulate a comprehensive understanding of the combined narratives.[31]

## FINDINGS

The first impression of the text revealed that experiencing stress-related exhaustion had a significant impact on the participants and their everyday lives. Paralysing fatigue, in combination with cognitive impairment, contributed to difficulties in taking part and engaging in life. This raised feelings of disability and powerlessness, which contributed to participants' experiences of losing their foothold in life. Thoughts about life's meaning arose, as did feelings of being lost and trapped. Life was perceived as indefinitely paused, with possibilities of change limited, no matter how much they struggled.

### Structural analysis
#### Loss of access to oneself

Experiencing stress-related exhaustion made participants feel they had lost access to themselves. This loss was expressed in three subthemes: constraints determine conditions, loss of self-recognition and deprivation of dignity (table 2).

#### Constraints determine conditions

The participants lost access to functions they previously had in everyday life. The obstacles they faced took over their abilities to act as they wished in life. Health problems such as pain, anxiety and sleep disturbance were common, and paralysing fatigue usually forced them to rest. The loss of vital functions made every chore a challenge, and these limitations contributed to feelings of disability and regression; sometimes even their orientation in time and space was affected.

> I am confused, so my family has helped me keep track of time, appointments, helped me pay the bills, and so on. (10)

Participants also described having difficulty holding a conversation because they lacked energy and had problems finding the right words.

#### Loss of self-recognition

Feeling a loss of control due to the constraints determining their everyday conditions contributed to a loss of self-recognition. Participants described their previous selves as able to manage a high pace or to handle several tasks at the same time—things that were no longer possible. The participants experienced a significant discrepancy between their previous and their actual selves, and this loss seemed to affect their personality and felt as if parts of themselves were missing.

> I have been able to work from four-thirty in the morning until nine p.m. After work, I arrived home and continued with household chores, for example, washing, cleaning, and other things. I used to have such a high capacity. (8)

This experience caused feelings of frustration and anger and feelings of shame and guilt towards their significant others. The noticeable difference in their appearance worried both themselves and people close to them.

> Many asked how I felt. I replied as you usually do: "I am tired". And there was a colleague who especially pushed me to seek help because I was not myself. (10)

| Table 2 | Subthemes, themes and main theme | | |
|---|---|---|---|
| **Main theme** | **Theme** | **Subtheme** | **Example of quotes for each theme** |
| Facing a blind alley | Loss of access to oneself | Constraints determine conditions | 'I am confused so my family has helped me keep track of time,appointments, helped me pay the bills, and so on' (10) |
| | | Loss of self-recognition | |
| | | Deprivation of dignity | |
| | Endless struggle | Managing everyday life | 'I live alone with a child who has a disability. I never sleep a whole night and have not done that in nine years, which leads to consequences. Here I am. I keep on doing everything without seeing any way out of this' (6) |
| | | Searching for explanations and understanding | |
| | | Process of negotiation | |
| | Lost in the middle of nowhere | Life on hold | 'My life is paused. It is on hold. Life goes on out there. My friends and other people's lives go on, but mine is in some limbo because nothing is going on. I exist, but there is no life in it. My studies are interrupted. Everything is interrupted, and I do not live. Living dead or…' (12) |
| | | Excluded from life | |
| | | Rethinking life | |

### Deprivation of dignity

Their reduced abilities to perform activities that had previously given them pride and self-esteem contributed to participants' experiences of losing dignity. Previous roles and positions, both private and professional, changed with the condition, and their earlier strength and courage decreased.

> I was sitting in the waiting room, feeling week, helpless, and useless. We justify our existence by being productive in some way, and one feels bad when one is not able to pay one's dues and needs to take up the physician's time. I felt worthless. (12)

Deprivation of dignity was especially evident with healthcare professionals, being considered a patient and treated as a sick person. Having stress-related exhaustion was an overwhelming experience, and not being cared for and treated with respect made the participants feel undignified.

> I lost my foothold and felt very weak when the physician expressed himself that way: 'If you do not plan to kill yourself, I don't need to take you on sick leave.' It was terrible! I have never identified myself as a person who gives up like that. (5)

### Endless struggle

The exemplars revealed that the participants experienced an endless struggle, which was expressed in the following three subthemes: managing everyday life, searching for explanations and understanding and process of negotiation (table 2).

### Managing everyday life

It was a challenging struggle to maintain the tempo necessary for coping with daily life. Life offered limited opportunities for change, and the participants saw no other options than to continue their struggle. They used strategies and compensated for their disabilities by reducing commitments both at work and in private life. Trying to act as they had before required a great effort, but they felt obliged to their family to do so.

> I live alone with a child who has a disability. I never sleep a whole night and have not done that in nine years, which leads to consequences. Here I am. I keep on doing everything without seeing any way out of this. (6)

### Searching for explanations and understanding

The participants searched for explanations and understanding about their condition and they struggled to find answers.

> I always try to reach answers to why this happened to me, but I do not know if I have reached something yet. Everything is just speculation, and I hope something is right, but I don't know. (12)

They were often referred between different physicians and healthcare facilities, which contributed to their increased frustration and stress. Repeated visits and answers that nothing was wrong, despite their feeling ill, left them feeling they were a burden, and they started to distrust their ability to understand themselves.

> In the end, I started thinking, Have I become mad? Have I misunderstood my signals? Should my life be like this for the rest of my life? (9)

### Process of negotiation

Various answers from healthcare professionals were under constant negotiation and involved participants rejecting, defending and trying to accept their condition.

> I experienced a big conspiracy against me when everyone had this mental approach, and I was sure that it was biological. All the bodily symptoms made me believe something physically was wrong with me. Then four physicians, who all had this mental approach, made me feel overpowered, and I had to let go of the hypothesis I'd hung onto. (12)

Accepting their limitations and the fact that they had been affected by stress-related exhaustion posed challenges, especially if their treatment by healthcare professionals was less supportive and involved challenging process of negotiation with themselves.

> The most difficult thing for me was to admit I was sick. It took me several days, or it took several weeks, actually. It was tough, but once I had admitted it to myself, had emailed my boss, had sent the message to the private healthcare insurance, then I released that burden and accepted the situation. (8)

### Lost in the middle of nowhere

The narratives also showed that participants felt lost in the middle of nowhere, which was expressed in the following three subthemes: life on hold, excluded from life and rethinking life (table 2).

### Life on hold

Other people's lives seemed to move ahead, leaving the participants behind, and they felt they were going around in circles with little ability to influence the situation. They felt emptiness, meaninglessness and a life with restricted content.

> My life is paused. It is on hold. Life goes on out there. My friends and other people's lives go on, but mine is in some limbo because nothing is going on. I exist, but there is no life in it. My studies are interrupted. Everything is interrupted, and I do not live. Living dead or… (12)

The fact that no time interval for recovery could be specified raised feelings of uncertainty and powerlessness. These feelings were a struggle to handle when searching

for motivation to continue the everyday fight. The advice from healthcare professionals was to rest, which was perceived as pointless because it had no noticeable effect on their recovery.

> I do not know what needs to happen or what to do to get better. I am resting all the time, but I notice no improvement. It is like trying to start a car without a battery. Pointless. The motor is broken, so it is no idea. You turn the key, but it is dead. So, I honestly do not know how to manage it or how someone should succeed in doing so. (11)

### *Excluded from life*

The participants felt excluded from essential parts of their lives such as work and spending time with family and friends. They described feeling excluded from their familiar, everyday context and felt uncertain about what the future would hold.

> I have somehow been excluded from my old life. It is as if I look at my past life through my memories. It is like a great villa. In this villa is my early life, but the door is locked, and I stand looking into a window and see everything I could do before. None of this I can do anymore. It doesn't work. (12)

Activities that previously gave life its meaning were no longer possible, which contributed to limiting the joyful elements in their everyday lives. Spending time on things that previously gave life its meaning led instead to grief and sadness.

> … He asked me if I had thoughts of killing myself, which I never reflected on before. However, when he asked me, I noticed that I had been thinking about it. Because it was the football World Cup and I couldn't watch TV, I couldn't eat, I couldn't do anything joyful. (8)

### *Rethinking life*

Having stress-related exhaustion made the participants reflect on their previous and present life situations. They reassessed previous actions and goals in life, and their prior way of living was questioned.

> I have hated lying on the couch all my life, and now it's the only thing I can do. It is such a rethinking of life. That you, in the end, are committed and obliged to do what you have never wanted to do. It's like the body says you don't get away with this, or there is no shortcut. (12)

Most of the participants felt responsible for what had happened. An emotional burden arose when they realised that their condition affected themselves and others. Now when facing the consequences, they felt disappointment to themselves and to others. Looking back could raise not only feelings of guilt and shame but also insights into what needed to change in their lives.

> For me, I see this basically as something positive. It does not feel positive, but I think it is a life lesson not to act as I have done so far. So, actually, the illness is a consequence of how I have lived my life, and the onset of the illness is a turning point, thus a chance to turn it over. (3)

### Interpretation of the whole

Having stress-related exhaustion was a life-changing experience for the participants and had a significant impact on their self-conception and their everyday lives. Their entire existence changed, and their selves and lives were perceived as unrecognisable. They struggled to regain access to themselves and a sense of meaning. This situation was interpreted as facing a blind alley, taking a fruitless and mistaken direction in life that forced them to stop, change direction and act differently. The blind alley, however, can represent both a crisis and an opening to new insights.

## DISCUSSION

In contrast to other studies,[37 38] the participants in our study were interviewed in the early stages of their sick leave. Our findings revealed that the participants struggled with a loss of vital functions, questioned their prior ways of life and faced a new life situation. This is consistent with findings from a study later in the rehabilitation phase,[37] and thus indicates that the condition has a great impact on people in both early and later stages of sick leave. Another study after several months of rehabilitation showed that life was experienced as living in darkness and struggling with the creation of meaning.[38] This indicates that symptom burden and challenges described in the present study may continue long term.

We interpreted a link between stress-related exhaustion to people's personal identities and existential challenges. The experience of *losing access to oneself* and to one's context affects the persons' entire view of themselves. 'The engine is broken, and it is impossible to start the car', as one participant expressed. The discrepancy between previous and present capacities contributed to *loss of self-recognition* and was linked to one's identity. The participants struggled not only with the challenge of understanding their condition but also with their need to understand themselves and their identities. This is in line with findings from previous studies, which have described the condition as a struggle with a threatening nothingness,[39] being disconnected from oneself[17] and a disconnection from the body and world.[37]

The impact of stress-related exhaustion on a person's identity also raises questions about life and its meaning. An understanding of the meaning of being affected by stress-related exhaustion can be understood as a *limit situation*, as described by the existential philosopher and psychiatrist Karl Jaspers (1883–1969). Limit situations are those such as death, struggling, suffering, guilt and existential

anxiety that remain unchanged. People do not perceive the limit situation until they confront it concretely and personally. Limit situations are like a wall we run into and then collapse, and we cannot avoid confronting them.[36] The main theme of our analysis, *facing a blind alley,* can be both a limit situation and a metaphor for how stress-related exhaustion affects a person's existence and raises existential challenges such as 'Who am I without my professional identity?' 'Who am I without the ability to perform?' and 'What is valuable in my life when I cannot live life as desired?'. A further interpretation is that the participants experienced *life on hold* and were *excluded from life* because relevance and meaning were missing. They were powerless in a situation they could not overcome, life became meaningless. This can be understood as expressing an existential feeling of unrelatedness. Such existential vulnerability is reminiscent of Jasper's metaphor of the broken mussel shell.[40] Throughout life, humans create sustainable foundations through their views of life, like building a house that creates conditions and protects against unpredictable experiences. In a limit situation, this sustainable foundation is threatened, and the person, much like a mussel without a shell, is homeless and unprotected from life's challenges (Jaspers,[40] p248).

Consistent with our findings, Jasper's metaphor points out the vulnerability of people who struggle to understand and explain their experience of stress-related exhaustion to themselves and others. This vulnerability, combined with a lack of respectful care and treatment from healthcare professionals, has a negative impact on the person's identity and self. Engebretsen *et al* describe how patients who seek care for stress-related exhaustion experience the encounter with the general practitioner as a battle and feel they are distrusted by others. This results in disconnection from their lifeworld which in turn triggers shame reactions.[17] In our study, this experience was interpreted as *deprivation of dignity* due both to the guilt and shame caused by their views of their condition and feeling unheard in their encounters with healthcare professionals. Shame has been described as an emotion of self-assessment that causes the person to feel anxiety at the thought of how he or she is seen and judged by others. Unsupportive encounters affect how the persons attribute values of themselves as human beings with subsequently increased distress.[16] Knapstad *et al* found that shame among people on sick leave might prolong absence.[41] The participants thus struggled to understand and explain their condition not only to themselves but also to others in light of their feelings of loss of self and context. Paul Ricoeur (1913-2005) distinguishes between two fundamental aspects of the self or identity,[42] which he calls *ipse* (selfhood) and *idem* (sameness). Ipse is that kind of inexpressible inner core that marks the core of who we are. Idem, on the other hand, is a more external way of identifying the self. Ipse identifies 'who' the self is and idem is 'what' the self consists of.[42] Our study findings indicate healthcare professionals focus on idem

(sameness) and disregard ipse (selfhood), which means they may not meet the person beyond the disorder. This raises concerns, especially as the condition has a significant impact on the person's identity. Hence, it is essential to know the person with stress-related exhaustion as a person first, as emphasised in person-centred care. Person-centred care takes the patient's narratives as the starting point for patients and healthcare professionals to cocreate individual plans of care and shared decision-making.[43] Carefully listening to patients' narratives is a core component to identify his/her resources and needs and to achieve a common understanding. Considering patients' narratives in the clinical encounter enables them to become an active part in finding solutions for health problems and strengthen their self-care abilities.[43 44]

Jaspers concept is useful to understand stress-related exhaustion as a limit situation, which is an experience of importance to humans, a life event that one cannot avoid.[36] Facing a blind alley implies that the participants are in a limit situation; they are forced to stop, rethink their decisions and find a new direction and a more suitable way in life. Jaspers' limit situation also entails an opening to deeper insights and therefore carries an inherent contradiction: one can experience and understand both one's limitations and one's possibilities. When the inner and outer contexts the participants built up collapse and a new direction in life is necessary, they start *rethinking life*. Earlier life choices and life goals are reassessed; they question their life situation, identity, value and meaning of life. *Rethinking life* seems to be the starting point to regain both a way forward in life and the belief that life is meaningful.

## Methodological limitations

The study included 12 participants' narratives, which could be argued do not to apply to other people's lives. The qualitative approach, however, aims not to reach generalisable findings, but to attain a deeper understanding of people's experiences; this study can, therefore, provide an understanding of people's lived experiences of stress-related exhaustion. The phenomenological hermeneutic approach aims to obtain understanding of people's individual perspectives via the interpretation of their narratives as texts.[31 33] To ensure trustworthiness, the qualitative framework for interpretative research by de Witt and Ploeg[45] was applied, which comprises of five characteristics: balanced integration, openness, concreteness, resonance and actualisation. Balanced integration was achieved by using a phenomenological hermeneutical approach where quotes from the participants were related to Karl Jaspers concept of limit situation. Openness was demonstrated by accounting for decisions that were made throughout the study, such as inclusion criteria, cessation of data collection and the analysis process. A reflexive process established the preconceived attitudes and beliefs to reduce biases in the process of data collection, analysis and interpretation. Regarding concreteness, the findings were presented by examples

of quotes that situate the reader concretely in the experience and contexts of being affected by stress-related exhaustion. Resonance was reproduced by the intensity and reflective nature of the participant's voice in their expression of the condition. For actualisation, the study results were not conclusive but could be interpreted by readers to deepen their understanding.

## CONCLUSION AND CLINICAL IMPLICATIONS

This study revealed that stress-related exhaustion is a challenging and life-changing experience that encompasses both a crisis and an opening towards new insights. When encountering patients with stress-related exhaustion, it is of importance to acknowledge the condition as severe and complex, which affects the person's identity and includes existential issues. Persons affected by stress-related exhaustion are often in urgent need of a caring dialogue to deal with feelings of shame, guilt and meaninglessness. Healthcare professionals are responsible for initiating this dialogue and for creating a respectful care environment that allows patients to narrate about their experiences and existential issues, which is a key to understanding the condition and finding new opportunities in their situation. Thoroughly listening to the patient's narrative enables to acquire knowledge about the patient, as a person, with needs, including wishes and feelings. This, together with the expertise of the healthcare professional, can be used to cocreate a personal health plan, also including realistic and achievable goals, to be followed up during recovery. One crucial aspect is also to identify and use the patient's resources and competences to make it possible for the person to see the crisis as a way to learn from it. Such a holistic approach, which takes into account the personal experience of the condition, may avoid deprivation of the person's dignity and can be implemented by applying person-centred care in clinical practice. Hence, we suggest combining the patients' and the healthcare professionals' perspectives in the clinical encounter, as a way to improve health and to optimise recovery for people with stress-related exhaustion.

**Contributors** All authors participated in the design of the study. SA conducted and transcribed the interviews. SA performed the analysis with critical input from LA, IE and AF. SA drafted the manuscript, which was reviewed and edited by LA, IE and AF. All authors approved the final manuscript. The authors wish to acknowledge the participants for sharing their experiences.

**Funding** This study was financed by grants from The Swedish Research Council for Health, Working Life and Welfare (reference number 2016–07418, 2017–00557 and 2019–01726) and from the Swedish state under the ALF agreement between the Swedish government and the country (ALFGBG-772191 and ALFGBG-932659). This work was also supported by The University of Gothenburg Centre for Person-Centred Care (GPCC), Sweden. GPCC is funded by the Swedish Government's grant for Strategic Research Areas (Care Sciences) and the University of Gothenburg, Sweden.

**Competing interests** None declared.

**Patient consent for publication** Not required.

**Ethics approval** Approval for the study was obtained from the Ethical Review Board at the University of Gothenburg (Dnr: 497–17, T 526–18) and the study conformed to the Declaration of Helsinki.

**Provenance and peer review** Not commissioned; externally peer reviewed.

**Data availability statement** No additional data are available.

**ORCID iDs**
Sara Alsén http://orcid.org/0000-0002-5691-3656
Lilas Ali http://orcid.org/0000-0001-7027-4371
Andreas Fors http://orcid.org/0000-0001-8980-0538

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
