## [Reviewer comments · BMJ Open]

ARTICLE DETAILS

TITLE (PROVISIONAL)	Facing a blind alley Experiences of stress-related exhaustion – a qualitative study
AUTHORS	Alsén, Sara; Ali, Lilas; Ekman, Inger; Fors, Andreas

VERSION 1 – REVIEW

REVIEWER	Maddi Olano Universidad de Navarra Spain
REVIEW RETURNED	16-Apr-2020

GENERAL COMMENTS	Thank you very much for giving me the opportunity to review this interesting article. It is a relevant and interesting study for health professionals. However, I believe that the authors need to make some changes to the document to be accepted for publication: - "Strengths and limitations of this study" section: The authors should further specify the strengths of the study, as they are very general. In addition, they should also point out the limitations.- Introduction: The relevance of the topic is adequately shown, however, this section should serve to justify the novelty and current interest of this topic and to do so, more current references should be used.- Method:* The authors should explain the conceptual framework that guided the study. In the discussion section they mention the philosopher Jasper, but it is not clear to what extent his theory could have influenced the results.* Sample: The authors should clarify if they achieved data saturation when they decided to complete the sample with 12 participants. It seems that it was not the case, as they explain the motives for not involving 7 other participants who were invited.* Inclusion criteria: It would be interesting for the authors to explain if they have taken into account the following inclusion criteria: Appropriate level of spoken and written language; Physical and mental capacity to conduct an interview; Time after diagnosis. This last criterion could be very relevant when interpreting the results.* Recruitment: Specify who did it.* It would be interesting to attach the complete guide of the interview.
---

	* Analysis: Although the authors cite Ricoeur and describe the phases of the analysis, it would be recommendable to add in table 2 examples of fragments of interviews from which the themes were derived. * The authors should specify the criteria of methodological rigour that was followed and justify its achievement. - Discussion: * As I pointed out before, Jasper's protagonism draws attention, explain his influence on the study. * In general, the number of studies with which the results are compared is somewhat scarce. Moreover, with some exceptions, the references are not very current. * The usefulness of patient narratives is mentioned several times. This is very interesting, but either in the methodology section or here, they should better specify what it means to use "narratives"; since they can be used in different ways and for different purposes. - Clinical implications: The authors should be more specific about which will be the direct implications of this study in practice. - References: Reference 10 is incomplete.
--	---

REVIEWER	Sue Monaro Concord Repatriation General Hospital Northern Sydney Local Health District The University of Sydney
REVIEW RETURNED	17-Apr-2020

GENERAL COMMENTS	The article presents qualitative findings very succinctly. I have made a number of comments in the manuscript to improve the readability and incorporate terminology for qualitative research. Many of these relate to fine-tuning the use of English language and being less tentative in reporting your findings. - The reviewer provided a marked copy with additional comments. Please contact the publisher for full details.
--

VERSION 1 – AUTHOR RESPONSE

Reviewer 1	Response
1) "Strengths and limitations of this study" section: The authors should further specify the strengths of the study, as they are very general. In addition, they should also point out the limitations.	Thank you. We have revised the strengths of the study and added limitations at page 2, heading: "strengths and limitations of this study".
2) Introduction: The relevance of the topic is adequately shown, however, this section should serve to justify the novelty and current interest of this topic and to do so, more current references should be used.	Thank you. We have added more up-to-date references at the beginning and at the end of the introduction (page 2-3).
3) - Method: The authors should explain the conceptual framework that guided the study.	Thank you for raising an important issue. In this paper we used a phenomenological hermeneutical

In the discussion section they mention the philosopher Jasper, but it is not clear to what extent his theory could have influenced the results.	method, which consists of three steps: naïve reading, structural analysis and interpretation of the whole (Lindseth and Norberg (A phenomenological hermeneutical method for researching lived experience. Scand J Caring Sci. 2004 Jun;18(2):145-53). The naïve reading and structural analysis (Table 2) is the first part of the interpretation process. The naïve reading can be seen as the phenomenological contribution, there we are trying to capture the intentionality in the narratives and be as open as possible to what the text want to say, while the structural analysis aims at viewing the text as objectively as possible. Meaning units are decontextualized from the text as a whole, i.e. the textparts are seen as independently as possible from their context in the text. As the reviewer points out it is important to consider the influence of Jasper´s framework, and as described by Lindseth and Norberg theoretical concepts are recommended to be included in the third step in order to enrich and deepen the understanding of the data. Hence, in this third step of the analysis we consulted the literature and considered Jaspers concept as appropriate. This is the reason why the concept of Jasper is introduced for the first time in the discussion section (as the interpretation of the whole is embedded under this heading). Jaspers framework has now been mentioned in the data analysis at page 4. At page 10 and 11 in the discussion and methodological limitations (page 11-12), we have clarified and elaborated on how the concept of “limit situation” was helpful to deepen the understanding of being affected by stress-related exhaustion.
4) Sample: The authors should clarify if they achieved data saturation when they decided to complete the sample with 12 participants. It seems that it was not the case, as they explain the motives for not involving 7 other participants who were invited.	Agree. We have added that the interviews were conducted until the authors considered the research question as fully answered (page 3, heading: Participants and setting).
5) Inclusion criteria: It would be interesting for the authors to explain if they have taken into account the following inclusion criteria: Appropriate level of spoken and written language; Physical and mental capacity to conduct an interview; Time after diagnosis. This last criterion could be very relevant when interpreting the results.	Thank you. Physical and mental capacity to participate was an inclusion criterion. Unfortunately, we missed to be clear about this but it has now been added at page 3. Restrictions in the Swedish language was not an exclusion criterion as we planned to use an interpreter if needed. As stated in the inclusion criteria (page 3), ongoing sick-leave, which had not exceeded 6 months was an inclusion criterion. Under the heading “data collection” at page 3 it is also mentioned that the interviews were performed 2-4 months after the sick leave had commenced. We agree that this is relevant when interpreting the results, and this is discussed in relation to findings from other studies in the second section of the discussion at page 9.
6) Recruitment: Specify who did it	Thank you for this notice. At page 3, under the

	heading: "Participants and setting" section, we have clarified that designated health care professionals screened patients on sick leave due to stress related exhaustion from medical records. After the patients have given their consent to participate, the first author contacted the patients by telephone to schedule a time for an interview.
7) It would be interesting to attach the complete guide of the interview.	Agree. A translated version of the interview guide has been added as a supplementary file.
8) Analysis: Although the authors cite Ricoeur and describe the phases of the analysis, it would be recommendable to add in table 2 examples of fragments of interviews from which the themes were derived.	In Table 2 (at page 5) we have added examples of quotes from the interviews.
9) The authors should specify the criteria of methodological rigour that was followed and justify its achievement.	We agree. At page 11-12 we have now added a section under the heading "methodological limitations section" to clarify how this study adhered to Witt & Ploeg's critical appraisal of rigour in interpretative research.
10) Discussion: - As I pointed out before, Jasper's protagonism draws attention, explain his influence on the study.	Please see our previous response to the third comment. At page 10 and 11 in the discussion, and methodological limitations (page 11-12), we have clarified and elaborated on how the concept of "limit situation" was helpful to deepen the understanding of being affected by stress-related exhaustion.
11) In general, the number of studies with which the results are compared is somewhat scarce. Moreover, with some exceptions, the references are not very current.	We have added text with up-to date references in the discussion at page 10.
12) The usefulness of patient narratives is mentioned several times. This is very interesting, but either in the methodology section or here, they should better specify what it means to use "narratives"; since they can be used in different ways and for different purposes.	Thank you for emphasizing this. We have elaborated on the benefits of using narratives at page 11.
- Clinical implications: The authors should be more specific about which will be the direct implications of this study in practice.	We have made amendments in this section and hope that the direct implications are better clarified.
- References: Reference 10 is incomplete.	Thank you noticing this, reference 10 has been corrected. Page 13
Reviewer 2	Response
Comments in the article and review of the language.	Thank you. We appreciate the reviewer's comments and editing of the language in the PDF file. Below the comments have been excerpted from the PDF and included in this document.
Abstract - Introduction needed	According to the author guidelines of BMJ Open an introduction in the abstract is not mandatory but it has been added at page 1.

Can you cite some WHO data?	References to WHO has been added at page 2.
Do you mean 'mental health'???	Thank you. We do think that clinical practice also covers mental health practice.
Percentage of sick leave taken?	Yes. We have added this information at page 3, under the heading “Method, participants and settings”.
Late introduction of Heideggerian concept. Maybe just keep it simple and delete habitual reverse order of headings	Thank you for this advice. The word “habitual”, at page 10 has been deleted. This is changed in Table 2 at page 5.
Leonard V (1989): A Heideggerian phenomenologic perspective on the concept of the person. Advances in Nursing Science 11, 40-55.	Thank you for this suggestion. We used the reference of Lindseth and Norberg (A phenomenological hermeneutical method for researching lived experience. Scand J Caring Sci. 2004 Jun;18(2):145-53, which has been added for this paragraph at page 4, in the “Data analysis” section.
Discuss rigour. See:de Witt L & Ploeg J (2006): Critical appraisal of rigour in interpretive phenomenological nursing research. Journal of Advanced Nursing 55, 215-229	Thank you for this suggestion. We have added a section under the heading “methodological limitation” at page 11-12 to clarify how Witt & Ploeg’s critical appraisal of rigour in interpretative research was applied.
Authors must include a statement in the methods section of the manuscript under the sub-heading 'Patient and Public Involvement'.	Thank you for noticing this. A statement in the method section under the heading “Patient and Public Involvement” has been added at page 4.
Patient advisers should also be thanked in the contributor ship statement/acknowledgements. If patients and or public were not involved please state this.	Good point. This has been added at page 12.

VERSION 2 – REVIEW

REVIEWER	Maddi Olano-Lizarraga Universidad de Navarra Spain
REVIEW RETURNED	27-May-2020

GENERAL COMMENTS	Congratulations to the authors for the work they have done. The changes they have made have undoubtedly improved the quality of the article. I suggest that the authors further specify the implications for practice. To this end, I advise them to provide concrete indications based on their results. There are still some incomplete references, as is the case with number 12.
---

REVIEWER	Sue Monaro Northern Sydney Local Health District
REVIEW RETURNED	Northern Sydney Local Health District 05-Jun-2020

VERSION 2 – AUTHOR RESPONSE

Reviewer #1	Response
I suggest that the authors further specify the implications for practice. To this end, I advise them to provide concrete indications based on their results.	Thank you for this comment. We agree and have carefully revised this section at page 12 in the hope that the direct implications now are better clarified.
There are still some incomplete references, as is the case with number 12.	Thank you noticing this, reference number 12 has been corrected.
Reviewer #2	
Overall a good revision with now just some deleting of excessive word as marked.	Thank you. We appreciate the reviewer's thorough review and editing of the language in the PDF file, and the revised version of the manuscript has been changed in line with the reviewer's suggestions. We have carefully considered the heading (Discussion/Interpretation of the whole) at page 9 and suggest to have separate headings. In order to be congruent with the description of the method, were interpretation of the whole is the final step of the analysis, we think that this part is important to emphasize. We have also added separate headings for the structured analysis. Page 5.

VERSION 3 – REVIEW

REVIEWER	Maddi Olano-Lizarraga Universidad de Navarra Spain
REVIEW RETURNED	Spain 07-Jul-2020

GENERAL COMMENTS	Many thanks to the authors for all the changes made. I think that the quality and clarity of the article is adequate for publication.
---

REVIEWER	Sue Monaro Concord Repatriation General Hospital Concord New South Wales Australia
REVIEW RETURNED	09-Jul-2020

GENERAL COMMENTS	The manuscript is reading well.
---------------------------------